# Obesity Management in the Primary Prevention of Hepatocellular Carcinoma

**DOI:** 10.3390/cancers14164051

**Published:** 2022-08-22

**Authors:** Elizabeth R. M. Zunica, Elizabeth C. Heintz, Christopher L. Axelrod, John P. Kirwan

**Affiliations:** Integrated Physiology and Molecular Medicine Laboratory, Pennington Biomedical Research Center, 6400 Perkins Road, Baton Rouge, LA 70808, USA

**Keywords:** obesity, weight loss, non-alcoholic fatty liver disease, non-alcoholic steatohepatitis, hepatocellular carcinoma, liver cancer, primary prevention

## Abstract

**Simple Summary:**

Hepatocellular carcinoma (HCC) is an increasingly prevalent and deadly disease driven in part by the rising obesity epidemic. Obesity causes HCC by initiating and progressing fatty liver disease. As such, weight reduction with the intention to treat fatty liver disease appears ideal for the primary prevention of HCC, but evidence is limited. This review covers recent advances in the treatment and management of obesity and fatty liver disease as it pertains to HCC risk and prevention. We conclude that combinatorial lifestyle, behavioral, medical, and surgical therapies that maximize whole body and liver fat reduction have the greatest potential to prevent HCC; however, prospective studies are required to demonstrate effectiveness.

**Abstract:**

Hepatocellular carcinoma (HCC) is the most frequent primary hepatic malignancy and a leading cause of cancer-related death globally. HCC is associated with an indolent clinical presentation, resulting in frequent advanced stage diagnoses where surgical resection or transplant therapies are not an option and medical therapies are largely ineffective at improving survival. As such, there is a critical need to identify and enhance primary prevention strategies to mitigate HCC-related morbidity and mortality. Obesity is an independent risk factor for the onset and progression of HCC. Furthermore, obesity is a leading cause of nonalcoholic steatohepatitis (NASH), the fasting growing etiological factor of HCC. Herein, we review evolving clinical and mechanistic associations between obesity and hepatocarcinogenesis with an emphasis on the therapeutic efficacy of prevailing lifestyle/behavioral, medical, and surgical treatment strategies for weight reduction and NASH reversal.

## 1. Introduction

The primary established risk factors for the development of HCC are biological sex, race/ethnicity, chronic alcohol consumption, viral infection, pre-existing liver disease, obesity, and type 2 diabetes [1]. HCC is not known to be caused by specific genetic mutations, but increased susceptibility is observed in patients with specific germline DNA polymorphisms including *EGF*, *IFNL3*, *MICA*, *TLL1*, *PNPLA3*, *TM6SF2*, and *MBOAT7* [2] as well as heritable conditions such as Wilson’s disease, tyrosinemia, alpha1-antitrypsin deficiency, and porphyria cutanea tarda [3]. The onset of HCC is frequently observed in patients with high alcohol intake and chronic viral hepatitis B or C infection, accounting for greater than 50% of cases [4]. Prevention of hepatitis-induced HCC occurs through prevention of exposure, inoculation with the hepatitis B vaccine (currently there is no hepatitis C vaccine), and/or treatment with targeted antiviral medications [5]. The expansion of vaccine inoculation in the United States, which was incorporated into the infant inoculation schedule in 1991, has greatly reduced the risk of hepatitis and cancer related mortality [6].

Independent of viral and toxic alcohol exposure, HCC is also strongly associated with the development of non-alcoholic fatty liver disease (NAFLD), defined as >5% steatosis in hepatocytes not caused by alcohol or viral injury. NAFLD initially presents as simple steatosis (NAFL) during which lipids accumulate as intrahepatic vesicles without affecting essential organ functions. In ~45% of patients, NAFL progresses to steatohepatitis (NASH), characterized by hepatocyte ballooning, apoptosis, accumulation of Mallory-Denk bodies, and inflammation within the liver parenchyma and/or portal vein. Additionally, scarring (fibrosis) can occur in response to persistent NASH and if left untreated, ultimately progress to irreversible fibrosis (cirrhosis) and liver dysfunction (decompensated cirrhosis). Specifically, NASH and cirrhosis are strongly associated with HCC incidence and mortality [7,8]. Notably, obesity is the primary driver of NAFLD [9], and approximately 50%, 80%, and 43% of patients with NAFL, NASH, and cirrhosis present with obesity, respectively [10]. Given the high prevalence of obesity in patients with NAFLD, the European Liver Patient’s Association (ELPA) proposed a change of nomenclature to metabolic dysfunction-associated fatty liver disease (MAFLD; used hereafter), defined as liver fat deposition along with obesity, diabetes, or combined metabolic disorders (Figure 1) [11]. Additionally, these guidelines suggest transitioning to the use of a graded continuum, identifying the degree of inflammation and fibrosis, to better classify severity and link to clinical outcomes [11].

Over the past 20 years, the incidence and prevalence of obesity, NAFLD, and obesity-related cancers, such as HCC, have risen dramatically [12]. Obesity is independently associated with ~2-fold risk of HCC onset [13], and patients with obesity exhibit a ~4-fold increase in HCC-related mortality and a ~2-fold increase in life-threatening complications following surgical cancer treatments [14]. Weight reduction dose-dependently improves biochemical markers of liver disease while decreasing the histological presence and activity of hepatosteatosis [15]. Recent clinical guidelines for the treatment of MAFLD indicate pragmatic dietary restriction as having more favorable outcomes but do not explicitly support intentional weight loss for disease management [16]. Furthermore, there is insufficient clinical evidence to support that weight loss prevents or restricts the onset of HCC. Herein, we review the evidence and discuss prospects for weight-reduction-oriented primary prevention of HCC with an emphasis on emerging lifestyle/behavioral, medical, and surgical therapies for obesity.

## 2. Epidemiology of HCC

Approximately one million patients are diagnosed annually with primary liver cancer, an ~2-fold increase from 1990 [17]. More than 80% of all primary liver cancers are histologically classified as HCC, 14.9% as intrahepatic cholangiocarcinoma, and the remaining 5.1% consist of other rare types such as angiosarcoma and hepatoblastoma [18]. Albeit regionally heterogenous on a global scale, the median age of HCC diagnosis is ~64 years of age [19]. Males are 4-fold more likely to develop HCC and account for ~75% of all cases [12]. The role of sex hormones is hypothesized to influence HCC development: estrogen is believed to be protective, whereas androgens have been observed to increase HCC, thus an increase of HCC incidence is observed in post-menopausal women [20]. Mortality is 2-fold greater in males compared to females, attributable to more severe disease staging at the time of diagnosis [12]. Importantly, after adjusting for age, women experience equivalent HCC morbidity and mortality for age- and stage-adjusted disease course [21,22]. Risk of NAFLD and progression of end-stage liver diseases are highly influenced by social determinants of health. For example, education and socioeconomic status, race, language barriers, social support, housing stability, food security, and health care access and literacy have all been associated with increased risk of onset and progression of fatty liver disease [23].

HCC predominantly manifests in the background of chronic liver disease but intertumoral and interpatient heterogeneity is high [24]. Cirrhosis of the liver is the most common underlying disease presentation, being present in >80% of all patients diagnosed with HCC. Cirrhosis describes the irreversible scarring of the liver that occurs through accumulated damage. Incidence of HCC in patients with cirrhosis is dramatically elevated by the presence of a precipitating factor, such as HBV (9-fold), HCV (7-fold), autoimmune hepatitis (6-fold), and/or NASH (45-fold) [25]. NAFL/NASH is predictive of mortality for both males and females [26]; obesity increases incidence, tumor size, and multiplicity in both female and male mice [27]. Furthermore, recent findings demonstrate that the prevalence of NASH is no longer dimorphic by sex [28] and that NASH is the fastest growing cause of HCC, having increased by ~40% from 2010 to 2019 [29]. HCC in non-cirrhotic livers typically present at more advanced stages, likely due to lack of routine surveillance which is recommended for people with cirrhosis. HCC in non-cirrhotic livers is often still in the background of liver disease, such as F3 fibrosis or viral infection and specifically. There is a strong association between MAFLD and HCC in non-cirrhotic livers [30,31]. Given the rise in MAFLD prevalence along with viral prevention and treatment strategies, MAFLD may soon be the predominant source of liver injury and HCC burden in both sexes. Thus, primary prevention and treatment of MAFLD are of critical importance for the prevention of HCC.

## 3. Obesity and HCC—A Convergence of Two Epidemics

Obesity is a chronic, relapsing, and progressive disease characterized by excess body fat. More than 700 million adults, or ~15% of all adults, were found to be living with obesity worldwide in 2020 and projected to reach over one billion adults in 2030 by the World Obesity Federation [32]. Obesity is most frequently diagnosed clinically according to body mass index (BMI), or proportion of body weight relative to the square of height (kg/m^2^). It is stratified into three classes based upon severity (class 1: 30–34.9, class 2: 35–39.9, class 3: ≥40 kg/m^2^) [33]. Notably, these cutoffs were identified to underestimate obesity-related health risks in Asian and South Asian populations, leading to a shifted scale for obesity for a BMI greater than 25 kg/m^2^ in these populations [34]. Obesity stems from chronic energy surplus triggered by the complex intersection of genetics, environment, lifestyle/behavior, psychology, co-morbidities, and certain medical therapies [35]. In addition to the physiological ramifications of obesity, ~50% of patients experience weight discrimination and bias, significantly reducing the quality of health care and utilization as well as increasing rates of mental health disorders such as depression and anxiety [36]. Obesity is an independent risk factor for the onset and progression of numerous conditions including type 2 diabetes, cardiovascular disease, hypertension, kidney disease, sleep apnea, and MAFLD [37]. More recently, obesity has been associated with an increased risk of at least 13 types of cancer, including meningioma, esophageal adenocarcinoma, multiple myeloma, renal, uterine, ovarian, thyroid, breast, gallbladder, stomach, pancreatic, colorectal, and liver cancer [38]. Collectively, these cancers account for over 40% of all diagnoses in the United States, several of which are increasing in severity [38]. The relationship between obesity and HCC is unique in that they share a known, direct linkage via MAFLD. Obesity is the primary cause of MAFLD. The prevalence of MAFLD mirrors that of obesity, which is estimated at ~30% of the population [39].

The etiology of HCC varies based upon the pathophysiological manifestation of the pre-existing liver disease. MAFLD can cause HCC directly from steatosis by predisposing hepatocytes to malignant transformation or through progression to steatohepatitis, fibrosis, and ultimately cirrhosis of the liver [7,8]. In the case of obesity, steatosis occurs when the rate of appearance of fatty acids exceeds the hepatic capacity for utilization and clearance, resulting in net retention in the form of lipid droplets, triglycerides, and/or cholesterol. There is consensus that some degree of hepatic fat is healthy but the precise diagnostic criteria delineating excess steatosis remains unclear. Currently, intrahepatic fat ≥5% of total tissue mass is the diagnostic threshold for hepatic steatosis [40]. However, such cutoffs are based on distributional normality within a population and not the point by which hepatic lipid accumulation adversely affects liver and/or peripheral metabolic health. This is evidenced in preclinical models where varying degrees of hepatic steatosis drive HCC independent of the progression to steatohepatitis, fibrosis, or cirrhosis [41]. Nonetheless, prolonged energy burden in the form of excess lipids induces progression to steatohepatitis, where macro- and micro-vesicular steatosis are observed in the presence of inflammatory infiltrates, and hepatocyte ballooning with or without fibrosis [42]. Importantly, some components of MAFLD, including steatohepatitis and early fibrosis, are reversible whereas cirrhosis represents a permanently scarred and damaged organ that inevitably progresses to failure [8]. The mechanistic progression of MAFLD to HCC is multifaceted and is still being elucidated. Current evidence indicates there is an alteration of immune cell function and response, sustained inflammation, an increase of mitochondrial reactive oxygen species (ROS), changes in cell cycle pathways such as activation of the PI3K/AKT/mTOR, and/or accumulation of genetic and epigenetic alterations, all of which drive hepatocyte DNA damage and oncogenic transformation [43].

Additionally, there are other pathophysiological features of obesity and metabolic syndrome, such as chronic low-grade inflammation [44] and dyslipidemia [45], that may contribute to the direct development of HCC independent of the progression to cirrhosis. The liver is responsible for both de novo cholesterol synthesis and uptake of excess cholesterol in the circulation. In the context of dyslipidemia, this can lead to an accumulation of oxidized LDL causing lipotoxicity and inflammation, promoting HCC [46]. Chronic intestinal inflammation compromises the gut barrier and alters microbiota populations, impacting intestinal metabolic function as well as that of other organs. Over the MAFLD continuum, a dysregulation of the enterohepatic axis forms, including increased intestinal permeability, altered bile acid and metabolite signaling, and imbalance of the gut microbiome, adding additional pathogenic layers to the progression of HCC [47]. Kupffer cells are the macrophages of the liver and mediators of liver injury and repair. Preclinical investigations demonstrate that Kupffer cells are activated by the proinflammatory state of obesity. They release cytokines, inducing inflammation and downregulation of insulin signaling driving MAFLD [48] and progression to HCC [44].

Currently, there is no FDA-approved drug for MAFLD. Thus, treatment is limited to lifestyle modifications primarily focused on obesity management. Routine care for patients with MAFLD varies among practitioners but minimally entails encouragement to reduce carbohydrate and fat intake and to increase exercise [49]. Specifically, the American Gastroenterological Association has outlined best practice advice statements highlighting that weight loss ≥5% of total body weight can decrease steatosis, ≥7% can resolve steatohepatitis, and ≥10% can decrease and/or stop the progression of fibrosis [49]. Medications that decrease body weight and blood glucose, prevent/scavenge free radicals, and stimulate bile acids are being investigated for MAFLD treatment [50,51].

## 4. Treatment Strategies for Obesity and Their Association with Improved Liver Health and HCC Prevention

### 4.1. Diet and Energy Balance

First-line therapy for obesity includes hypocaloric nutritional and behavioral intervention coupled with increasing physical activity, with current guidelines recommending about a 20–30% calorie restriction [52,53]. For patients with diagnosed MAFLD, specific limitation of energy-dense, nutrient-poor foods high in saturated and trans fats and simple sugars and avoidance of alcohol are advised, given the evidence of their effect independent of calorie intake [54,55,56]. There are a limited number of controlled studies assessing the impact of specific micro and/or macronutrients, such as vitamins D and E, carbohydrates, dietary fats, and proteins, on the progression of fatty liver [57]. Thus, most of the clinical recommendations focus on dietary patterns rather than specific dietary components. Evidence supporting the relationship between specific dietary patterns and HCC is limited to large observational prospective cohort studies [58,59,60,61,62,63,64,65,66,67,68,69,70,71,72,73,74] and subsequent meta-analyses [75,76,77,78,79,80,81]. Such approaches limit the ability to parse out the combined impacts of obesity, smoking, and alcohol intake and increasing physical activity and plant-centered dietary patterns. Promisingly, changes in dietary patterns to increase plant-based foods can decrease the risk of liver cancer development by ~30–50%, although there is little information with regards to effects on HCC recurrence and mortality [82]. Detailed in the World Cancer Research Fund’s (WCRF) most recent report on diet, physical activity, and cancer, there is strong evidence that overweight or obesity, consumption of alcohol, and consuming foods contaminated with aflatoxins increases the incidence of liver cancer [83]. There is also supporting evidence that coffee intake, physical activity, and consumption of fish decrease the risk of liver cancer [83].

Lifestyle interventions typically produce 5–10% weight loss, but have high propensity for relapse and weight regain [84], limiting long-term efficacy, and cancer risk reduction. Alternative restrictions of calorie intake, such as alternate-day fasting and time-restricted eating, have gained interest in recent years for the potential for improved efficacy in the treatment of obesity [85,86], but the impact on liver disease and HCC remains preliminary. In pre-clinical models, induction of a chronic negative energy balance via calorie restriction [87,88], intermittent fasting [89], and/or time-restricted feeding [90] decreases the incidence and multiplicity of liver cancer. Mechanistically, prevention appears to be driven by metabolic improvements, the resolution of pre-existing liver disease, and the restoration of immune function [88,89,91]. Additionally, dietary intake impacts the composition of the gut microbiome, the type and concentration of circulating bile acids, and the integrity of the gut barrier permeability, all of which affect HCC development [91]. Overall, there is a need for improved investigation using the gold-standard controlled feeding studies to assess specific dietary interventions for the prevention of HCC and whether they require weight loss.

### 4.2. Physical Activity

Physical activity to achieve and maintain >5% weight loss is a first-line recommendation for treatment of early and intermediate stages of MAFLD [92,93]. The general recommendation for exercise as a treatment for MAFLD is 150 min/week of moderate-intensity exercise [94], but increasing duration and/or intensity leads to greater improvements in disease biomarkers [95]. While aerobic exercise induces superior benefit on MAFLD-related fibrosis when compared with resistance exercise [96,97], both aerobic and resistance exercise effectively reduce MAFLD through different intensities and oxygen consumption [16]. Additionally, resistance training is effective at sustaining muscle mass and quality in patients with chronic diseases, including liver disease [98,99]. Thus, current guidelines for patients with obesity and moderate to advanced liver disease recommend a combination of aerobic and resistance exercise [92,97].

HCC inversely correlates with the level of physical activity independent of other cardiometabolic risk factors, with risk of liver-related mortality decreasing as physical activity rises in a dose-dependent manner [100,101,102]. Furthermore, exercise reduces the incidence of HCC in mice with obesity and type 2 diabetes independent of changes in body weight, providing evidence for exercise as a direct preventative measure of HCC onset [103]. Therefore, physical activity presents as an effective preventative strategy for HCC and other liver diseases, particularly in patients with obesity.

Exercise is indicated as a treatment strategy for MAFLD via induction of weight loss, although the benefits of exercise in liver disease extend beyond weight control alone [104]. While exercise acutely increases hepatic glucose output due to increased energy demand, improvements in hepatic insulin sensitivity and suppression of hepatic glucose production are chronic adaptations to regular physical activity that ultimately contribute to increased rates of hepatic lipid oxidation and decreased hepatic lipid content [105,106]. Hepatic adaptations to exercise are regulated, in part, via activation of (AMP-activated protein kinase) AMPK and Unc-51-like autophagy activating kinase 1 (ULK1) signaling. AMPK/ULK1 signaling induces lipophagy, which subsequently reduces hepatic steatosis in MAFLD [107]. Additionally, proteins released by skeletal muscle during exercise, such as IL-6 and myonectin, improve hepatic metabolism [108,109]. Furthermore, proteins secreted from the liver during exercise, most notably angiopoietin-like 4 (*ANGPTL4*), lead to improvements in whole-body metabolic function via increased insulin sensitivity and lipid metabolism [110]. Physical activity is also correlated with improvements in hepatocarcinogenic signaling in patients with MAFLD, specifically *PEG10* [111]. Although studies on exercise as a treatment for hepatic cirrhosis are limited, there is evidence suggesting that exercise improves cirrhosis-driven sarcopenia and other whole-body metabolic parameters that could contribute to improvements in liver health [112]. Hence, physical activity conveys health benefits, particularly in cases of obesity, which prevent liver disease and HCC via activation of proteins released by skeletal muscle and liver that serve as potential therapeutic targets for treatment.

### 4.3. Pharmacotherapy

Patients with moderate to severe obesity or mild obesity with resistance to lifestyle therapies are considered for pharmacotherapy. Notably, only ~2% of all patients with obesity initiate medical therapy [113]. To date, there are five medical therapies approved for long-term use in the treatment of obesity including orlistat, phentermine/topiramate, naltrexone-bupropion, liraglutide, and semaglutide [114]. There is also a sixth approved drug, setmelanotide, for use in those with a specific rare genetic disorder. Four drugs are approved, phentermine, benzphetamine, diethylpropion, and phendimetrazine, for short-term (12 weeks) use to suppress appetite [114]. Though not FDA-approved for weight loss, metformin is also commonly prescribed off-label to manage body weight and improve metabolic function. Currently, pharmacological weight loss is generally comparable to lifestyle therapy with heterogeneity in responsiveness and slightly more durable efficacy; thus, some of these drugs are being investigated in clinical trials for MAFLD patients (Table 1) [51]. Other drugs and therapeutic targets, such as regulating energy expenditure, gut hormones, and microbiota, as well as genetic therapies, are being investigated for improved and sustained efficacy of weight loss [114]. Recently, tirzepatide, a dual-agonist drug approved for the treatment of type 2 diabetes, has demonstrated great promise as a potential treatment for obesity, resulting in a more than 20% weight reduction in a phase 3 double-blind, randomized controlled trial [115].

Abbreviations: AST, aspartate aminotransferase; ALT, alanine aminotransferase; GGT, gamma-glutamyl transferase; GLP-1, glucagon-like peptide 1; FXR, farnesoid X receptor; PPAR, peroxisome proliferator-activated receptor; GCGR, glucagon receptor; GIP-1, glucose-dependent insulinotropic polypeptide; SGLT2, sodium-glucose cotransporter-2; and TSH, thyroid stimulating hormone.

Orlistat is a gastrointestinal lipase inhibitor that modestly decreases body weight by restricting exogenous fat absorption. Orlistat treatment decreases MAFLD in patients with obesity, with little to no apparent benefit in reducing inflammation and fibrosis [144]. Orlistat was associated with acute liver toxicity in a subset of patients, which may limit application for cancer prevention [145]. The combination weight-reduction therapies phentermine/topiramate and naltrexone-bupropion have yet to be directly tested for prevention benefit to HCC. However, they are likely to have clinical value in that any degree of weight loss is associated with reduction of intrahepatic lipid. Liraglutide is a glucagon-like peptide-1 receptor agonist (GLP-1RA) that decreases body weight by reducing food intake. Liraglutide prevents the progression of MAFLD to HCC in mice with obesity and streptozotocin-induced diabetes [146]. A phase 2 clinical trial demonstrated resolution of steatohepatitis and attenuation of fibrotic progression with liraglutide in nearly 40% of participants [118]. Semaglutide, another GLP1RA used in obesity treatment that can induce up to 20% weight loss [119], reduces NASH in patients with advanced fibrosis [120]. Notably, there remains a debate whether human hepatocytes express GLP-1 receptors and thus whether the mechanism of action for GLP-1RA’s are direct or indirect. Thus much of the potential benefits are attributed to the weight-lowering effects [147]. Metformin is associated with ~70% reduced risk of developing HCC in patients with obesity and type 2 diabetes [148]. Mechanistically, metformin has modest but long-term weight loss effects [121] and appears to attenuate HCC development by improving hepatic function, restricting stellate cell activation, decreasing steatosis and fibrosis, and halting decompensation of cirrhosis in mice [149]. However, such observations have yet to be confirmed in humans where the disease pathogenesis and background are more complex.

Notably, other therapeutic candidates being investigated for the treatment of MAFLD, such as antioxidants, farnesoid X receptor (FXR) agonists, peroxisome proliferator-activated receptor (PPAR) agonists, anti-diabetic drugs, and dyslipidemia drugs, have varying effects on body weight (Table 1) [150,151]. For example, several lead drugs undergoing phase 2 and phase 3 clinical trials, such as FXR agonists, display a significant decrease in body weight [152]. Others, such as the thyroid hormone receptor-beta agonist resmetirom, have no effect on body weight [143]. Conversely, PPAR agonists, such as Lanifibranor, demonstrate potential to resolve steatohepatitis and improve fibrosis [131], but often result in some weight gain limiting their therapeutic potential. A promising therapeutic approach is the development of drugs that target steatosis, hepatitis, and fibrosis through multiple modes of action, such GLP-1, GCGR, and/or GIP-1. Many of these drugs are effective in reducing body weight and have demonstrated preclinical efficacy in the treatment of MAFLD and are now in phase 1 and 2 clinical trials [153]. Overall, there is limited data available from both pre-clinical models and humans to conclude whether pharmacological treatment of obesity prevents HCC, but newer obesity medications display promise as they pertain to cancer prevention.

### 4.4. Metabolic Surgery

For patients with moderate to severe obesity and metabolic risk factors, bariatric/metabolic surgeries (BMS), such as vertical sleeve gastrectomy (VSG) or Roux-en-Y gastric bypass (RYGB), are used for weight management and comorbidity treatment. BMS aims to achieve significant, sustainable weight loss and improvement and/or resolution of related diseases, such as obstructive sleep apnea (96% remission rate), type 2 diabetes (92% remission rate), dyslipidemia (76% remission rate), hypertension (75% remission rate), and cardiovascular disease (58% remission rate) [154]. Weight loss is more substantial than lifestyle or medical therapy, commonly exceeding 25% loss of body weight or more for 10 years or longer after treatment with infrequent relapse [155]. Currently, the most utilized BMS procedures are the VSG, consisting of ~85% exclusion of the stomach curvature, and RYGB, consisting of combined exclusion of 60–70% stomach and 30% small intestine. Other BMS procedures include gastric banding (an adjustable band wrapped around the upper part of the stomach to restrict food intake) and duodenal switch (exclusion of 60–70% stomach and 75% small intestine). Additionally, endoscopic procedures for the treatment of obesity have been developed, such as sleeve gastroplasty (exclusion from inside the stomach) and intragastric balloon (fluid or gas-filled balloon placed into the stomach to restrict food intake) [156]. BMS, specifically VSG and RYGB, lowers body weight [157,158], resolves MAFLD [159,160,161], and normalizes hepatic function [162,163] in humans and rodents. Promisingly some endoscopic weight-reducing procedures have also been shown to induce >10 *v*/*v*% weight loss with improvements in liver function [164,165], but did not reduce cirrhosis-related complications such as HCC in a small pilot study [166]. Prior BMS is associated with lower incidence [167,168,169], prevalence [170], and mortality [171] of HCC. However, utilization of BMS for obesity treatment is poor, with less than 1% of eligible patients receiving treatment [172], and to date, there is no prospective evidence that it prevents the onset and progression of HCC.

The mechanisms of BMS and improved liver health are not yet fully elucidated. Potential mechanisms include an increase in gut hormone signaling such as GLP-1, remission of hepatic and adipose tissue inflammation, lipid malabsorption, and augmentation of enterohepatic bile acid secretion and signaling [173,174]. BMS decreases food intake and adiposity [157,175], which are independent predictors of onset and progression of NAFLD/NASH to HCC [15]. Importantly, VSG and RYGB markedly alter gastrointestinal secretions and inter-organ communication prior to significant weight loss [176], which may independently alter hepatocarcinogenesis. These findings indicate potential weight-independent mechanisms for HCC prevention. However, there is no direct human or rodent evidence or mechanistic understanding of how BMS lowers the risk of obesity-driven HCC, ultimately limiting clinical application as a prevention strategy. Thus, it remains unclear if and/or how BMS confers protection from HCC beyond weight reduction [15] and whether those effects are procedure-specific [177]. Importantly, RYGB and VSG appear similarly effective in treating NAFLD/NASH [178,179], but RYGB is generally superior for sustaining weight loss and metabolic health benefits [159,177], which may differentially impact HCC incidence.

## 5. Conclusions

Obesity is a highly prevalent and recurrent disease that impacts risk of HCC by exacerbating the onset and progression of MAFLD. Consistently, sustained weight loss decreases HCC risk by delaying or reversing the progression of MAFLD based on the magnitude of body fat reduction. Obesity treatment strategies that decrease body weight by 10% or more have the greatest impact on early to intermediate stage MAFLD. To date, emerging pharmacological and surgical approaches for weight reduction have not been evaluated for primary prevention of HCC nor has there been an adequate comparison of the modality of weight loss for the prevention of HCC. Furthermore, while there may be differences in magnitude of weight loss and long-term efficacy between approaches, there is an overall lack of practice and continuity of treatment in the management of obesity which limits comparative evaluation. Combinatorial lifestyle, pharmaceutical, and/or surgical therapies (Figure 2) are most likely to yield superior long-term HCC prevention by conferring durable and sustained weight reduction that can be modified to patient response to treatment. Furthermore, improved mechanistic understanding of the impact of diet, physical activity, and BMS on liver health may offer new insight regarding therapeutic targets for the treatment of MAFLD and prevention of HCC.

## Figures and Tables

**Figure 1 cancers-14-04051-f001:**
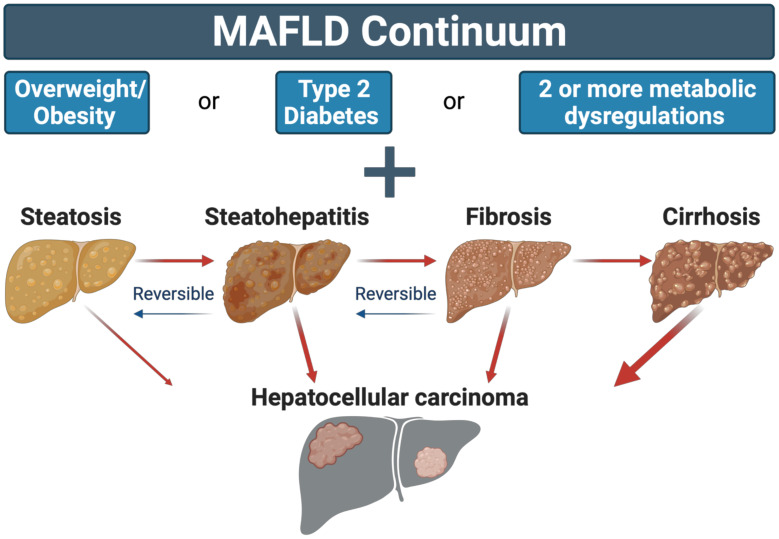
The MAFLD continuum characterizes a spectrum of disease initiated by hepatic lipid accumulation observed in combination with overweight or obesity, type 2 diabetes, or two or more metabolic dysregulations. Across the spectrum, varying severity of MAFLD can independently lead to hepatocellular carcinoma.

**Figure 2 cancers-14-04051-f002:**
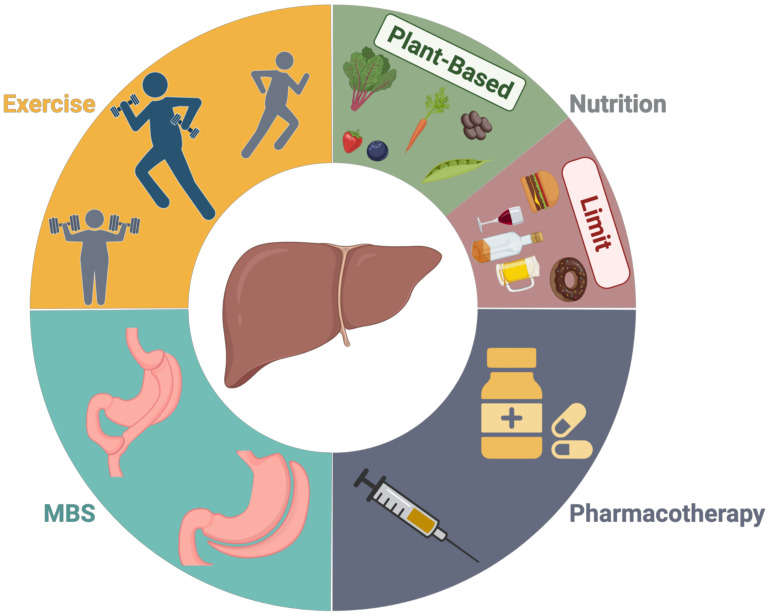
A combination approach for effective-sustained weight reduction is recommended for preventing obesity-related hepatocellular carcinoma (HCC). A combination of lifestyle, surgical, and/or pharmacotherapy interventions allows for the development of a person-centered adaptive approach to weight reduction and weight maintenance. Both aerobic and resistance exercise have been found to reverse the severity of MAFLD, albeit via different mechanisms. Thus, a combination of aerobic and resistance exercise is recommended for improved liver health, maintenance of lean body mass, and 5% weight reduction. Evolving expert panel consensus has determined that dietary patterns that support increased plant-based consumption and limit energy-dense, nutrient-poor foods and alcohol decrease the risk of HCC. For people who have MAFLD, a 20–30% calorie restriction is recommended to achieve 5–10% weight reduction, with specific limitations of saturated and trans fats and simple sugars. BMS can induce upwards of 25–50% sustained weight reduction and has been shown to reverse the severity of MAFLD and reduce the risk of HCC. Weight-reducing pharmacotherapies hold great potential for the treatment of MAFLD and have been shown to reduce the risk of HCC.

**Table 1 cancers-14-04051-t001:** Pharmacological Approaches to the Treatment of Obesity-related MAFLD.

Drug Name	Weight Effect	Clinical Evidence for the Treatment of MAFLD
**Lipase inhibitors**
Orlistat *	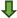	[116]	Reduced AST, ALT, GGT and no change in fibrosis [117]
**Dual Anorectic and Anticonvulsants**
Phentermine-topiramate *	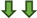	[116]	None found
**Dual Norepinephrine–Dopamine Reuptake Inhibitor and Opiate Antagonists**
Bupropion-naltrexone *	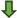	[116]	None found
**GLP-1 agonists**
Liraglutide *	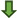	[116]	Resolution of steatohepatitis and decreased fibrosis [118]
Semaglutide *		[119]	Resolution of steatohepatitis [120]
**Biguanides**
Metformin	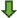	[121]	No difference in steatosis, hepatitis, or fibrosis [122]
**FXR agonists**
Ursodeoxycholic acid	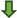	[123]	Can reduce serum ALT and GGT [124]No significant effects on liver histology [124]
Obeticholic acid	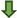	[125]	Decreased fibrosis [126] Significant elevation in low-density lipoprotein cholesterol and reduction in high-density lipoprotein cholesterol [127]
Cilofexor	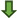	[128]	No significant effects on liver histology
**PPAR agonists**
Pioglitazone	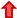	[122]	Improvement in steatosis, hepatitis, and ballooning, no change in fibrosis [122,129]
Elafibranor	No change [130]	Resolution of steatohepatitis [130]
Lanifibranor	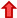	[131]	Resolution of steatohepatitis and decreased perisinusoidal fibrosis [131]
Saroglitazar	No change [132,133]	Reduced ALT and liver fat content [132]Reduced steatosis and ballooning [134]
**Dual GLP-1/GCGR agonists**
Cotadutide	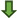	[135]	Reduced AST, ALT, GGT, steatosis and fibrosis indices [135]
**Dual GLP-1/GIP-1 agonists**
Tirzepatide		[115]	Reduction in liver fat content [136]
**SGLT2 inhibitors**
Canagliflozin	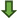	[137]	Reduced ALT, AST, fibrosis index [138,139]Reduced AST, ALT, GGT, and fibrosis index [140]
Dapagliflozin	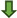	[137]	Reduced ALT, AST, GGT [139,141]
Empagliflozin	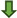	[137]	Decreased steatosis, ballooning, fibrosis [139,142]
**TSH β agonists**
Resmetirom	No change [143]	Reduction and resolution of steatosis, reduction in ballooning and inflammation, and markers of liver injury and fibrosis [143]

* FDA approved for weight loss as of July 2022. 
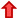
 Increase in body weight. 
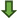
 Low decrease in body weight (5–9%). 
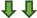
 Moderate decrease in body weight (10–15%). 

 High decrease in body weight (16–20%). 

 Very-high decrease in body weight (>20%).

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
