# Peer review of "Obesity Management in the Primary Prevention of Hepatocellular Carcinoma"

_cancers, 2022, doi:10.3390/cancers14164051_

Round 1

Reviewer 1 Report

In this article, the authors aimed to review the clinical and mechanistic associations between obesity and hepatocarcinogenesis.

Although there are already several reviews on this topic, it remains an area of ​​great importance in clinical practice. The article reviews current aspects of obesity management in the prevention of hepatocellular carcinoma, namely behavioral, medical and surgical treatment strategies.

However, the authors have barely addressed endoscopic options in the treatment of obesity. It would be interesting, for example, to review the association between intragastric balloon placement and decreased risk of hepatocellular carcinoma.

Author Response

Response to Reviewer 1 Comments

Point 1: In this article, the authors aimed to review the clinical and mechanistic associations between obesity and hepatocarcinogenesis. Although there are already several reviews on this topic, it remains an area of ​​great importance in clinical practice. The article reviews current aspects of obesity management in the prevention of hepatocellular carcinoma, namely behavioral, medical, and surgical treatment strategies.

Response 1: We thank the reviewer for their time and thoughtful consideration of our work.

Point 2: However, the authors have barely addressed endoscopic options in the treatment of obesity. It would be interesting, for example, to review the association between intragastric balloon placement and decreased risk of hepatocellular carcinoma.

Response 2: We thank the reviewer for this comment and have since modified the review to include additional information on endoscopic obesity treatments. To our knowledge, there is minimal evidence that endoscopic treatments specifically are associated with reduced HCC risk. However, weight loss with certain procedures such as sleeve gastroplasty has been associated with improvements in glycemic control, hepatic enzymes, body composition, and circulating lipids, all of which may be indicative of advancing liver disease.

Reviewer 2 Report

This review describes the management of obesity as a primary prevention of HCC.

Obesity is a leading cause of HCC, one of the leading causes of cancer-related deaths worldwide. Prospects for primary treatment of HCC by managing obesity through diet, exercise, medical therapy, and surgery are summarized.

It may be useful to readers as a status report on the possibility of primary prevention of HCC through management of obesity.

Major problem

Many of the descriptions are unclear as to whether each of the treatments was actually capable of primary prevention of liver cancer. In the management of obesity, the biggest problem is the lack of practice and continuity of treatment, even if there is a method, and this point is not mentioned.

Author Response

Response to Reviewer 2 Comments

Point 1: This review describes the management of obesity as a primary prevention of HCC. Obesity is a leading cause of HCC, one of the leading causes of cancer-related deaths worldwide. Prospects for primary treatment of HCC by managing obesity through diet, exercise, medical therapy, and surgery are summarized. It may be useful to readers as a status report on the possibility of primary prevention of HCC through management of obesity.

 Response 1: We thank the reviewer for considering our work related to primary prevention of HCC.

Point 2: Many of the descriptions are unclear as to whether each of the treatments was actually capable of primary prevention of liver cancer. In the management of obesity, the biggest problem is the lack of practice and continuity of treatment, even if there is a method, and this point is not mentioned.

Response 2: We acknolwedge the concern of the reviewer and have since updated the manuscript to clarify the relationship(s) between obesity treatment and liver cancer endpoints. At this time, there is little direct, prospective evidence demonstrating that sustained weight reduction prevents liver cancer. However, there is evidence, as we present, that intermediate endpoints that dictate disease onset such as NAFLD and NASH are favorably altered by weight reduction strategies to varying degrees. It is our belief that cancer prevention will likely involve combinatorial strategies that maximize NAFLD/NASH treatment to prevent progression to HCC.